# A Mathematical Approach Using Strat-M^®^ to Predict the Percutaneous Absorption of Chemicals under Finite Dose Conditions

**DOI:** 10.3390/pharmaceutics14071370

**Published:** 2022-06-28

**Authors:** Ryoki Kunita, Takafumi Nishijima, Hiroaki Todo, Kenji Sugibayashi, Hitoshi Sakaguchi

**Affiliations:** 1Safety Science Research Laboratories, Kao Corporation, Kanagawa 250-0002, Japan; nishijima.takafumi@kao.com (T.N.); sakaguchi.hitoshi@kao.com (H.S.); 2Faculty of Pharmacy and Pharmaceutical Sciences, Josai University, Saitama 350-0295, Japan; ht-todo@josai.ac.jp (H.T.); sugib@josai.ac.jp (K.S.); 3Faculty of Pharmaceutical Sciences, Josai International University, Chiba 283-8885, Japan

**Keywords:** percutaneous absorption, artificial skin membrane, Fick’s first law of diffusion, finite dose condition, safety assessment

## Abstract

Estimation of the percutaneous absorption is essential for the safety assessment of cosmetic and dermopharmaceutical products. Currently, an artificial membrane, Strat-M^®^, has been focused on as the tool which could obtain the permeation parameters close to the skin-derived values. Nevertheless, few practical methodologies using the permeation parameters for assessing percutaneous absorption under in-use conditions are available. In the present study, based on Fick’s first law of diffusion, a novel mathematical model incorporating the permeation parameters as well as considering the water evaporation (*T_eva_*) was constructed. Then, to evaluate the applicability domain of our model in the case where Strat-M^®^-derived parameters were used, the permeation parameters were compared between the skin from edible porcine and Strat-M^®^. Regarding chemicals (−0.2 ≤ Log *K_ow_* ≤ 2.0), their permeation profiles were equivalent between Strat-M^®^ and porcine skin. Therefore, for these chemicals, the percutaneous absorption was calculated using our model with the permeation parameters obtained using Strat-M^®^ and the *T_eva_* determined by measuring the solution weight. The calculated values revealed a good correlation to the values obtained using porcine skin in finite dose experiments, suggesting that our mathematical approach with Strat-M^®^ would be useful for the future safety assessment of cosmetic and dermopharmaceutical products.

## 1. Introduction

In the safety assessment of cosmetic and dermopharmaceutical products, it is necessary to investigate percutaneous absorption, which represents the cumulative amount permeating through the stratum corneum (SC), when chemicals are applied with finite doses under in-use conditions. Indeed, the main exposure to cosmetic products is from dermal pathways [1]. In vitro skin permeation experiments using human or animal skin are generally used to evaluate percutaneous absorption [2]. Although in vitro experiments using skin are generally expected to reflect in vivo performances, the use of skin tissue presents disadvantages that include its high cost and low reproducibility. Nowadays, the development of evaluation methods not requiring the use of skin has attracted attention, while artificial membranes or reconstructed cultured human skin models have been proposed and investigated as alternatives to human or animal skin worldwide [3].

Among these tools, Strat-M^®^ (EMD Millipore corp., Burlington, MA, USA) is an artificial membrane that mimics the skin tissue and consists of two layers of polyethersulfone (PES) containing artificial synthetic lipids and one layer of polyolefin. Comparative studies on the permeability of chemicals (i.e., permeability coefficient, *Kp*) between Strat-M^®^ and the human skin revealed that their relationships were highly correlated [4,5]. In addition, Strat-M^®^ exhibited a permeation profile which was closer to the skin compared to other artificial membranes (i.e., dialysis tubing cellulose, silicone) [6]. Based on these properties, Strat-M^®^ is currently attracting attention as an alternative membrane to skin for in vitro permeation experiments.

Physicochemical properties such as the molecular weight (*MW*) and apparent distribution ratios of *n*-octanol/water (Log *K_ow_*) of chemicals were shown to have significant effects on their permeability [7,8]. Consequently, the *Kp* can be further divided into two parameters: diffusion parameter (*DL*^−2^) and partition parameter (*KL*), where *D*, *K* and *L* indicate diffusion coefficient, partition coefficient and skin thickness, respectively [9]. These permeation parameters can be easily calculated in in vitro permeation experiments using skin, but also alternative membranes such as Strat-M^®^. Measuring these parameters with Strat-M^®^ would be useful to unravel the permeation characteristics of various chemicals. Furthermore, measurements of chemical permeation with Strat-M^®^ showing a high reproducibility were also reported [10].

While the use of Strat-M^®^ has various advantages, some issues remain. For instance, the amount of transdermal absorbed chemical is normally calculated from the amount of chemical permeated through the skin and measured in the viable epidermis/dermis layers (VED). Assessing the amount of chemical penetrating in the VED, however, cannot be estimated from experiments with Strat-M^®^ due to the difficulty of separating each layer.

In addition, permeation parameters such as *Kp*, *K*, and *D* can be determined from the steady-state of flux, without considering solvent evaporation from the applied formulation [11]. However, the effect of evaporation cannot be ignored in finite dose experiments that reflect in-use conditions. If each chemical concentration in the applied solution and/or the composition of the applied solution were to change due to evaporation of volatile substances, the permeation rate would be also modified [12]. It is thus difficult when using Start-M^®^ to evaluate the percutaneous absorption under in-use conditions, as required for safety assessment.

Buist et al. have proposed a mathematical model including *Kp* and *K*, based on Fick’s first law of diffusion, to predict the percutaneous absorption under finite dose conditions [13]. We thought that application of the permeation parameters obtained using Strat-M^®^ to this mathematical model might make the percutaneous absorption under finite dose conditions predictable. However, they have shown that the predicted values were overestimated when their model was applied to aqueous solutions. For an aqueous solution, due to solvent evaporation, the chemical would precipitate within 24 h and keep not penetrating [14]. Nevertheless, in the prediction model described above, a steady percutaneous absorption was assumed for 24 h. This assumption may explain the overestimation of predicted values for the percutaneous absorption of chemicals from aqueous solutions. Therefore, to predict percutaneous absorption under finite dose conditions accurately, it appears necessary to consider water evaporation from the applied solution.

In the present study, a novel mathematical model incorporating the permeation parameters (*Kp* and *K*) as well as considering water evaporation (*T_eva_*) was developed to evaluate percutaneous absorption under finite dose conditions. Furthermore, to assess the scope of application of our mathematical model, where the parameters derived using Strat-M^®^ instead of the skin were applied, the permeation parameters between porcine skin and Strat-M^®^ were compared. These experiments were conducted using various chemicals (Log *K_ow_*: −2.3 to 2.0).

## 2. Materials and Methods

### 2.1. Materials

Kojic acid (KA), caffeine (CAF), *p*-hydroxy benzoate (PHBA), citric acid, trisodium citrate, methanol, acetonitrile (high-performance liquid chromatography (HPLC) grade), and phosphoric acid were purchased from Fujifilm Wako Pure Chemicals Industries, Ltd. (Osaka, Japan). Sodium benzoate (BA) was acquired from Fushimi Pharmaceutical Co., Ltd. (Kagawa, Japan). 5-Isosorbide mononitrate (ISMN) was purchased from Tokyo Chemical Industries, Ltd. (Tokyo, Japan). Lidocaine (LID) was acquired from Combi-Blocks, Inc. (San Diego, CA, USA). Methylparaben (MP) was obtained from Sharon Laboratories (Ashdod, Israel), while pH 7.0 phosphate-buffered saline (PBS) was purchased from Thermo Fisher Scientific K.K. (Kanagawa, Japan).

To prepare citric acid buffer (pH 3.0 and pH 5.0) as solvents, 1.1% (*w*/*v*) or 0.39% (*w*/*v*) citric acid and 0.35% (*w*/*v*) or 0.87% (*w*/*v*) trisodium citrate were mixed with pure water. In parallel, KA, CAF, ISMN and MP were dissolved in double-distilled water to reach 1.0% (*w*/*v*) or saturated solubility. Sodium benzoate was dissolved in PBS or pH 3.0 citric acid buffer to reach 1.0% (*w*/*v*) or saturated solubility. Lidocaine was dissolved in pH 5.0 citric acid buffer to reach 1.0% (*w*/*v*). Table 1 summarises the physicochemical properties and concentration of chemicals in aqueous solution.

### 2.2. In Vitro Skin Permeation Experiment under Infinite Dose Conditions

The experimental method was compliant with the *OECD test guideline 428* [2]. Frozen porcine ears (Tokyo Shibaura Zoki K.K., Tokyo, Japan) were thawed and rinsed with tap water. The skin was excised, and its thickness measured with a micrometre (Mitsutoyo corp., Kanagawa, Japan). Hairs were subsequently trimmed and shaved. Skin with a thickness of 1 mm or less was used for the experiments. The excised skin was applied to the Franz diffusion cell (effective diffusion area of 1.0 cm^2^; PermeGear Inc., Hellertown, PA, USA). When the sample solvent was water, the receiver compartment was filled with 8.0 mL of PBS. Otherwise, it was filled with 8.0 mL of the same solvent as the samples. To acclimate the skin to the experimental condition, the Franz diffusion cell was placed in an incubator (temperature 32 ± 1 °C, humidity 50 ± 2%; Tokyo Science Instruments, Tokyo, Japan) for 1 h. Afterwards, trans-epidermal water loss (TEWL) was measured using a Vapo metre (Delfin Technologies Ltd., Kuopio, Finland) to evaluate the integrity of the excised skin. Skin with a measured TEWL value of 15 g/m^2^/h or less [15] was used for the experiments. For Strat-M^®^ experiments, the membrane was applied to the Franz diffusion cell and similarly placed in the incubator. The receiver compartment was filled with 8.0 mL of pure water when the sample solvent was also water. Next, each solution was applied onto the skin or Strat-M^®^ with an infinite dose (500 µL/cm^2^). The opening was covered with aluminium foil (Mitsubishi Aluminum Co., Ltd., Tokyo, Japan) to prevent evaporation of water from the donor side. Every 2 h until 8 h after the onset of the experiment, an aliquot (500 µL) was withdrawn from the receiver solvent and the same volume of receiver solvent was added to the chamber to keep the volume constant.

### 2.3. In Vitro Percutaneous Absorption Experiment under Finite Dose Conditions

Using procedures described in Section 2.2., excised skin was applied to the Franz diffusion cell. Each chemical solution was then applied onto the skin with a finite dose (10 µL/cm^2^). All receiver solvent and skin were collected 24 h after the application. The skin was thoroughly washed with 2.0 mL of water. Then, the application sites were stripped up to 10 times, until the epidermis peeled off, using tapes (Clinical and Derm LLC, Dallas, TX, USA) to remove the SC. Targets were extracted from skin tissues using 20% (*v*/*v*) methanol. The samples were subsequently filtered using a polytetrafluoroethylene syringe filter (EMD Millipore corp.) and the targets quantified using HPLC.

### 2.4. Water Evaporation Experiment

The bottom of the glass cap lid in the Franz diffusion cell was covered with Parafilm^®^ (Pechiney Plastic Packaging Inc., Chicago, IL, USA) instead of skin or Strat-M^®^, and each solution (10 µL) was applied. This glass cap lid was placed in an incubator under the conditions described in Section 2.2. The weight of applied solution was measured every 0.25 h, until no change of weight was observed.

### 2.5. Quantification Method Using HPLC

Each sample was inserted into the HPLC system (Nexera-i LC-2040C plus, Shimadzu Co., Kyoto, Japan), with a volume of 20 μL for and ISMN, or 10 μL for the other chemicals besides MP. The YMC-Triart C18 column (4.6 mm × 250 mm, 5.0 μm; YMC Co., Ltd., Kyoto, Japan) was maintained at 40 °C. The flow rate was 1.0 mL/min. The conditions for the mobile phase and absorption wavelength of each target were as follows: 0.1% (*w*/*v*) phosphoric acid (acetonitrile = 95:5, 270 nm for KA; acetonitrile = 80:20, 270 nm for CAF; acetonitrile = 85:15, 230 nm for ISMN; acetonitrile = 70:30, 220 nm for LID; acetonitrile = 80:20, 230 nm for BA). Additionally, for MP and *p*-hydroxy benzoate (PHBA), which is the metabolite of MP, the insertion volume was 3.0 μL. The YMC-Triart C18 column (2.1 mm × 100 mm, 1.9 μm; YMC Co., Ltd.) was also maintained at 40 °C. The flow rate was 0.15 mL/min. The conditions for the mobile phase and absorption wavelength were as follows: 0.1% (*w*/*v*) phosphoric acid (acetonitrile = 85:15, 255 nm).

### 2.6. Theoretical

#### 2.6.1. Determination of Permeation Parameters

Permeation parameters (*Kp*, *K*, and *D*) were calculated from the time profile of the cumulative amount of chemical that permeated through the skin or Strat-M^®^ according to the following equations [10]:(1)$$Flux=\frac{KDCv}{L}=KpCv$$
(2)$$D=\frac{L^{2}}{{6T}_{lag}}$$
(3)$$K=\frac{{6T}_{lag}Kp}{L}$$
where *Flux* is the amount of chemical permeated through the membrane per unit of time and membrane area, *Cv* is the concentration of chemical, *L* is the thickness of membrane, and *T_la_*_g_ is a lag time calculated from the intersection of *x*-axis and the slope of steady flow rate in the permeation profile. In the present paper, skin thickness (*L*) = 0.10 (cm) and Strat-M^®^ thickness (*L*) = 0.03 (cm) were applied to obtain parameters for the mathematical model.

#### 2.6.2. Prediction of the Chemical Amount Permeated through the SC until the Complete Evaporation Time for the Solvent (*M*_1_)

Overall flow regarding percutaneous absorption under in-use conditions is described in Figure 1. When an aqueous solution is applied onto the skin surface, water is evaporated from each solution with passing time (0 ≤ *t* ≤ *T_eva_*, where *T_eva_* is the complete evaporation time for the solvent). Based on Fick’s first law of diffusion, the time-dependent chemical amount at the skin surface (*M_sur_* (*t*)) (0 ≤ *t* ≤ *T_eva_*) is expressed by following equation:(4)$$-\frac{{dM}_{sur}}{dt}{=KpC}_{sur}A$$
where *t*, *C_sur_* and *A* represent the time from application, the chemical concentration at the surface and the effective permeation area, respectively. With assumed solvent and *SC* as one compartment, the concentration of chemical in this compartment (*C_sur+SC_* (*t*)) was expressed using the chemical amount in this compartment (*M_sur+SC_* (*t*)) by the following equation:(5)$$C_{sur+SC}\left(t\right)=\frac{M_{sur+SC}\;(t)}{{V+V}_{SC}}$$
where *V* and *V_sc_* represent the solvent volume and the *SC* volume, respectively. According to Equations (4) and (5), the time-dependent *M_sur+SC_* (*t*) (0 ≤ *t* ≤ *T_eva_*) is expressed by the following equations:$$-\frac{{dM}_{sur+SC}}{dt}{=KpC}_{sur+SC}A$$
(6)$$-\frac{{dM}_{sur+SC}}{dt}=Kp\frac{M_{sur+SC}}{{V+V}_{SC}}A$$

Indeed, according to the evaporation process, the solvent volume will be decreased. In other words, *V* is also dependent on the time (*V* (*t*)). However, in this paper, *V* was assumed as a constant until passing over *T_evp_* to simplify the equations.

The cumulative chemical amount measured in the *SC* and at the skin surface over time until *T_eva_* (*M_sur+SC,_* (*T_eva_*)) is expressed by integrating Equation (6):$$\int_{M_{sur+SC}\;(0)}^{M_{sur+SC}{\;(T}_{eva})}-\frac{1}{M_{sur+SC}}{dM}_{sur+SC}=\int_{0}^{T_{eva}}Kp\frac{A}{{V+V}_{SC}}dt$$
(7)$$M_{sur+SC}\;\left(T_{eva}\right){=M}_{sur+SC}\;\left(0\right)exp\left(-\frac{{KpAT}_{eva}}{{V+V}_{SC}}\right)$$
where *M_sur+SC_* (0) represents the amount of chemical immediately after the application of solution on the skin surface. The amount of chemical that permeated through the *SC* over time until *T_eva_* after the application of solution (*M*_1_) is expressed by the difference between *M_sur+SC_* (0) and *M_sur+SC_* (*T_eva_*):(8)$$M_{1}{=M}_{sur+SC}\;(0)\;-{\;M}_{sur+SC}{\;(T}_{eva}{)=M}_{sur+SC}\;(0)\left(1-exp\left(-\frac{{KpAT}_{eva}}{{V+V}_{SC}}\right)\right)$$

#### 2.6.3. Prediction of the Chemical Amount Permeated through the SC from *T_eva_* to 24 h (*M*_2_)

After passing over *T_eva_*, the chemical on the skin surface does not keep penetrating the skin due to the solvent evaporation, but the chemical in the *SC* does migrate into the VED. In the case where the solvent volume was assumed as a constant at *T_evp_*, the chemical amount in the *SC* at *T_eva_* (*M_SC_* (*T_eva_*)) is expressed using *M_sur+SC_* (*T_eva_*) and Equation (7) by following equation:$$M_{SC}{\;(T}_{eva}{)=M}_{sur+SC}{\;(T}_{eva})\frac{V_{SC}K}{{V+V}_{SC}K}\;$$
(9)$$M_{SC}\;\left(T_{eva}\right){=M}_{sur+SC}\;\left(0\right)exp\left(-\frac{{KpAT}_{eva}}{{V+V}_{SC}}\right)\frac{V_{SC}K}{{V+V}_{SC}K}$$
where *K* represents the partition coefficient for solvent and *SC*. According to Equation (4), the time-dependent chemical amount in the *SC* (*M_SC_* (*t*)) (*T_eva_* < *t* ≤ 24) is expressed in the following equations:(10)$$-\frac{{dM}_{SC}}{dt}=Kp\frac{M_{SC}}{V_{SC}}A$$

The chemical amount in the *SC* measured 24 h after the application of solution (*M_SC_* (24)) is expressed by integrating Equation (10):$$\int_{M_{SC}{\;(T}_{eva})}^{M_{SC}\;(24)}-\frac{1}{M_{SC}}{dM}_{SC}=\int_{T_{eva}}^{24}Kp\frac{A}{V_{SC}}dt$$
(11)$${\;M}_{SC}{\;(24)=M}_{SC}{\;(T}_{eva})exp\left(-\;\frac{{KpA(24\;-\;T}_{eva})}{V_{SC}}\right)\;$$

The cumulative chemical amount permeated through the *SC* between *T_eva_* and 24 h (*M*_2_) is expressed by the difference between *M_SC_* (*T_eva_*) and *M_SC_* (24)_:_$$M_{2}{=M}_{SC}\;\left(T_{eva}\right)\;{-\;M}_{SC}\;\left(24\right){=M}_{SC}{\;(T}_{eva})\left(1-exp\left(-\;\frac{KpA\left({24\;-\;T}_{eva}\right)}{V_{SC}}\right)\right)$$

According to Equation (7), *M*_2_ could be expressed by Equation (12)
(12)$$M_{2}=\frac{M_{sur+SC}\;\left(0\right)V_{SC}K}{{V+V}_{SC}K}exp\left(-\frac{{KpAT}_{eva}}{{V+V}_{SC}}\right)\left(1-exp\left(-\;\frac{KpA\left({24\;-\;T}_{eva}\right)}{V_{SC}}\right)\right)$$

The cumulative amount of chemical that permeated through the *SC* until 24 h after the application (=percutaneous absorption) is expressed by a sum of *M*_1_ (Equation (8)) and *M*_2_ (Equation (12)).

In our research, *M_sur+SC_* (0) = 0.01·*Cv* (µg/cm^2^), application area (*A*) = 1.0 (cm^2^), solvent volume (*V*) = 0.01 mL (cm^3^), and *SC* volume (*V_sc_*) = 0.002 mL (cm^3^) were applied to Equations (8) and (12). In addition, *Kp*, *K*, and *T_eva_* obtained by each experiment as described in Section 2 and Equations (1)–(3) were also applied to these equations.

### 2.7. Statistical Analysis

The relationships between predicted and observed values were analysed with the Pearson’s correlation coefficient using Microsoft Excel 365. Statistical significance was to 5% (*p* values were <0.05) in all evaluations.

## 3. Results

### 3.1. Permeation Parameters Obtained from Infinite Dose Experiments

Figure 2a shows the permeation profiles through porcine skin after application of KA aqueous solution under an infinite dose. All data points measured from 2 h to 8 h show linearity. Skin permeation parameters were obtained as shown in Section 2.6.1. Unlike porcine skin, only the 2 h point was off the straight line in Strat-M^®^ (Figure 2b). In the case of Strat-M^®^, the permeation parameters were calculated from three data points between 4 h and 8 h. Table 2 summarises the obtained parameters from the permeation profiles.

### 3.2. Relationship in Permeation Parameters between Porcine Skin and Strat-M^®^

As shown in Figure 3a, the lipophilic chemicals, BA (pH 3.0) and MP, exhibited higher *Kp* values compared to other chemicals in the porcine skin. This result is in accordance with our findings using Strat-M^®^. On the other hand, with respect to hydrophilic chemicals (Log *K_ow_ <* −0.2), the relevance of *Kp* values to Log *K_ow_* differed between the porcine skin and Strat-M^®^ preparations. In the porcine skin, their *Kp* values were almost similar among hydrophilic chemicals, whereas in Strat-M^®^, the lower their Log *K*_ow_ values were, the lower their *Kp* values were. This result indicates that for hydrophilic chemicals it was harder to penetrate Strat-M^®^ than the porcine skin. In the case where *K* values were compared to Log *K_ow_* values, their relationships could be represented by a logistic curve in porcine skin and a linearity in Strat-M^®^, as shown in the relationship between *Kp* values and Log *K*_ow_ values (Figure 3b).

### 3.3. Solvent Evaporation from the Applied Solution

The evaporation kinetics from the aqueous solutions was measured when covering the glass cap lid with Parafilm^®^ instead of excised skin. Because the skin additionally contained some water before the start of the experiment, it might be difficult to assess the water evaporation accurately without skin. Figure 4 shows the evaporation kinetics from each examined solution. When the evaporation time reached 1.5 h, the change of weight was negligibility minimal for the different aqueous solutions. Thus, we set *T_eva_* to 1.5 h for the 10-µL aqueous solutions.

### 3.4. Application of a Mathematical Model for Predicting Percutaneous Absorption

To evaluate the validity of our mathematical model, percutaneous absorption ratios (percutaneous absorption divided by *M_sur+SC_* (0)) were predicted using the equations described in Section 2.6. and compared to observed values obtained by the experiment described in Section 2.3. Independent of the lipophilicity of applied chemicals, the predicted values were consistent with the observed ones, resulting in a high correlation between them (*r* = 0.876, *p* < 0.05) (Figure 5). This result reveals that our mathematical model was effective for predicting percutaneous absorption under in-use conditions.

### 3.5. Prediction of Percutaneous Absorption Using Strat-M^®^

As described in Section 3.2, the permeation of hydrophilic chemicals (KA, LID (pH 5.0) and BA (pH 7.0)) was significantly decreased in Strat-M^®^ compared to porcine skin. Therefore, with respect to four chemicals (CAF, ISMN, BA (pH 3.0), and MP), the parameters obtained using Strat-M^®^ were applied to our mathematical model and percutaneous absorption under finite dose conditions was estimated. Figure 6 shows the relationship between predicted values and observed values using porcine skin in finite dose experiments. The almost 1:1 relationship (slope ≈ 1.0) and the very high correlation coefficient (0.993, *p* < 0.05), between these values, were obtained among four chemicals.

## 4. Discussion

When evaluating percutaneous absorption for safety assessments, in vitro skin permeation experiments represent a general standard worldwide. Although skin provides a useful tool to understand chemical permeation, some issues are faced when using skin tissues (i.e., cost, time, reproducibility) [5]. In the present study, we focused on using the artificial membrane, Strat-M^®^, to simplify in vitro experiments. However, methodologies using Strat-M^®^ for evaluating percutaneous absorption under in-use conditions were not yet established. Therefore, we developed a novel evaluation method for assessing percutaneous absorption by combining Strat-M^®^ experiments with a mathematical approach. In particular, the Fick’s first law of diffusion incorporating water evaporation was applied. Several researchers have reported using Fick’s second law of diffusion with the evaporation rate of solvent water [12,16,17,18]. These approaches allowed to estimate the percutaneous absorption of chemicals, although many parameters such as drug diffusivity in the VED (*D_ved_*) and chemical partition coefficient into the VED (*K_ved_*) would be necessary in addition to the ones used in the present study. Besides, parameters corresponding to *D_ved_* and *K_ved_* could not be obtained from Strat-M^®^ due to its lack of physical division into several layers. Thus, developing this approach with the modified Fick’s first law of diffusion would be useful to estimate percutaneous absorption of chemicals under in-use conditions.

The skin permeation of chemicals is divided into two routes: the trans-SC comprising intercellular and intracellular pathways, and trans-appendage pathway such as hair and sweat glands [19]. Although the appendage area accounts for only approximately 0.1% of the entire skin [20], hydrophilic chemicals have been shown to pass through the appendages to reach the skin [21]. Additionally, regarding the reconstructed cultured human skin model, without an appendage route, a lower permeation of hydrophilic chemicals was confirmed than the skin [22]. Therefore, when interpreting our result in which low permeability of hydrophilic chemicals through Strat-M^®^ compared to porcine skin, we should consider that skin permeation of hydrophilic chemicals is inadequate in Strat-M^®^, similar to a reconstructed cultured human skin model.

When comparing our data with the published data [4], as shown in Figure 7, our results for four chemicals (CAF, ISMN, BA (pH 3.0), and MP) had a similar tendency to the results reported by Uchida and co-authors. This suggests a concordance of the *Kp* values for chemicals with Log *K_ow_* of −0.2 to 2.0 between porcine skin and Strat-M^®^. Furthermore, since the permeation parameters are essential to calculate the percutaneous absorption in our mathematical approach, a strong correlation of the permeation parameters between porcine skin and Strat-M^®^ would contribute to the proper estimation of the percutaneous absorption of chemicals (−0.2 ≤ Log *K_ow_* ≤ 2.0) in finite doses.

However, they confirmed a linear correlation of *Kp* values between Strat-M^®^ and human skin for various targets including hydrophilic chemicals, unlike our research. This difference might depend on the density of hair follicles and the size of hair follicle orifice: the porcine skin has higher density and large hair follicle orifice compared to the human skin [23]. Thus, hydrophilic chemicals are anticipated to permeate more easily through porcine skin than human skin. According to these findings, when our study is performed using human skin, not porcine skin, the obtained result may indicate that our approach using Strat-M^®^ is available for chemicals with Log *K_ow_* of −0.9 to 3.5. To clarify the applicability domain of our method, especially with regard to hydrophilic chemicals (Log *K_ow_* < −0.9), further experiments using human skin should be performed.

Concerning our mathematical model, several points should be noted. In this study, no consideration was taken for the decrease in solvent volume and the increase in chemical concentration to simplify the equation. Furthermore, changes in skin permeation parameters such as partition coefficient would occur when a complex formulation is applied due to a change in saturated concentration by evaporation of volatile components in the formulation [12]. In addition, the drug diffusion/partition coefficients are modified with the destruction of SC packaging structure, the extraction of SC lipids and the reduction in thermodynamic activity by other components in formulations such as alcohols, oils, and surfactants [24,25,26,27,28]. These factors would be also affected on Strat-M^®^. More recently, Arce et al. reported that the CAF permeation through Strat-M^®^ increased depending on the polyol content. This result was supposed due to the disruption of the integrity of Strat-M^®^ barrier [29]. Thereby, improvement of this model should be done so that it could be predictable for skin permeation of chemicals following the application of complex formulations in the future.

Currently, in silico approaches using quantitative structural activity relationship (QSAR) are becoming popular in the field of percutaneous absorption. Researchers have developed promising models allowing to predict percutaneous absorption without conducting experiments [30,31,32,33,34,35]. However, it should be noted that a few QSAR models are applicable to complex formulations under finite dose conditions [36], due to the difficulty of considering multiple factors for percutaneous absorption by calculation only. Therefore, in the present study, the hybrid method that combines a mathematical model and in vitro experiments using an artificial membrane was applied to resolve this issue. The obtained method might be considered more useful than in silico approaches for predicting the percutaneous absorption of chemicals in complex formulations under finite dose conditions.

## 5. Conclusions

In summary, we developed a novel mathematical model incorporating permeation and evaporation parameters (*Kp*, *K*, and *T_eva_*) to predict percutaneous absorption under in-use conditions. Our mathematical model, where calculated *Kp* values from porcine skin were used as a parameter, revealed a high predictability in percutaneous absorption under finite dose conditions. On the other hand, in the case where parameters derived using Strat-M^®^ were applied to our model, in a bounded scope (−0.2 ≤ Log *K_ow_* ≤ 2.0), a good correlation between predicted and observed values was observed. Although the results of the present study suggest the usefulness of our novel method using Strat-M^®^ in safety assessments, further experiments performed on various chemicals and formulations are warranted to clarify the scope of application of our method. The application of Strat-M^®^ as an alternative to skin has many advantages (cost-effective, easy to handle, high reproducibility and high-throughput). Therefore, the simultaneous analysis of multiple samples for screening would be also possible. In the future, our methodology is expected to contribute to formulation developments as well as safety assessments.

## Figures and Tables

**Figure 1 pharmaceutics-14-01370-f001:**
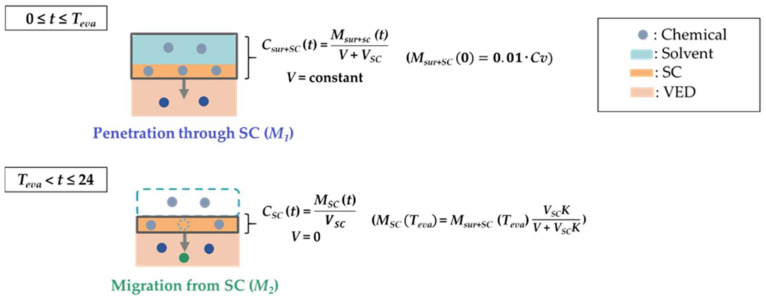
Flow diagram for time course of the chemical amount (*M* (*t*)) and concentration (*C* (*t*)) under finite dose conditions. *M_s_**_ur+SC_* (*t*) and *C_sur+SC_* (*t*) refer to the chemical amount and concentration at the skin surface and in the SC; *M_SC_* (*t*) and *C_SC_* (*t*) refer to the chemical amount and concentration in the SC; *V*, *Vsc* and *K* refer to solvent volume, SC volume and partition coefficient, respectively; *t* and *T_eva_* refer to time after starting the permeation experiment and time when the solvent was completely evaporated solvent, respectively.

**Figure 2 pharmaceutics-14-01370-f002:**
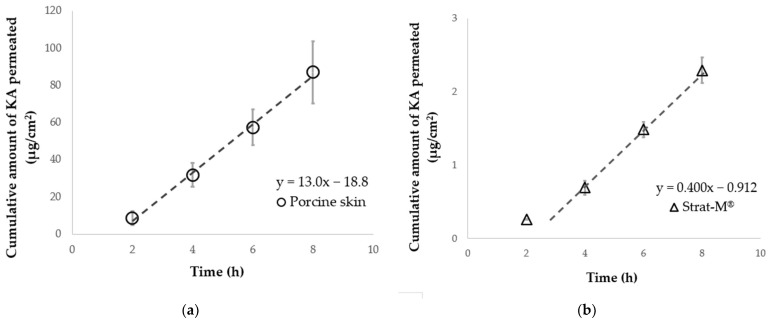
Time course of the cumulative amount of KA permeated under infinite dose conditions. Permeation profiles of KA applied through porcine skin (**a**) or Strat-M^®^ (**b**). Dashed lines represent the regression line and each point represents the mean ± standard error (S.E.) (*n* = 3).

**Figure 3 pharmaceutics-14-01370-f003:**
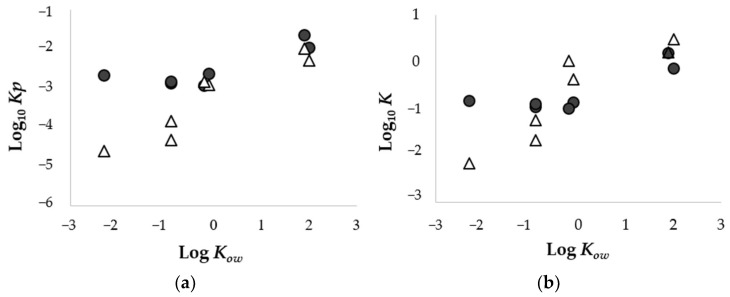
Relationship between the Log *K_ow_* and *Kp* (**a**) or *K* (**b**) of chemicals applied through porcine skin or Strat-M^®^. Circles (●) represent porcine skin and triangles (Δ) represent Strat-M^®^; each point represents the mean (*n* = 3−4).

**Figure 4 pharmaceutics-14-01370-f004:**
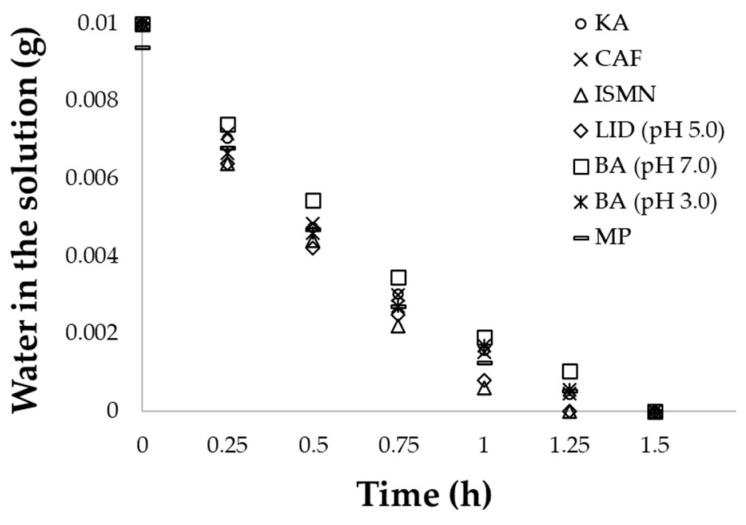
Time course of water evaporation from each aqueous solution. Each point represents an observed value (*n* = 1).

**Figure 5 pharmaceutics-14-01370-f005:**
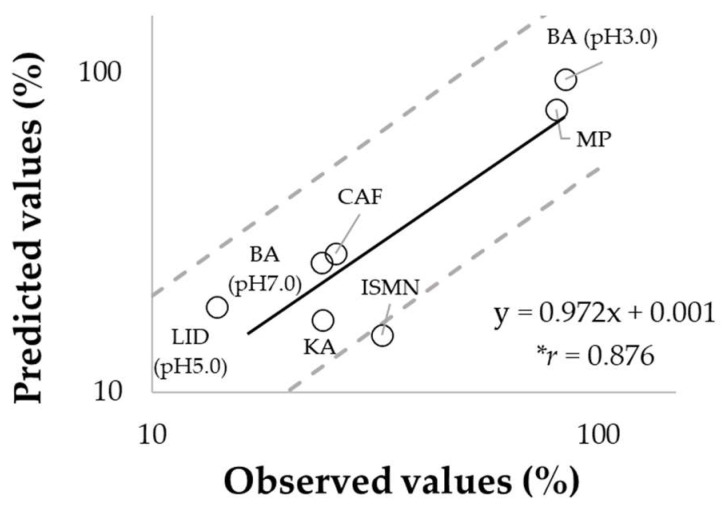
Comparison of predicted versus observed values using porcine skin for percutaneous absorption ratio. Each point represents the mean, while the dashed line corresponds to a factor 2. The Pearson’s correlation coefficient showed statistical significance (* *p* < 0.05).

**Figure 6 pharmaceutics-14-01370-f006:**
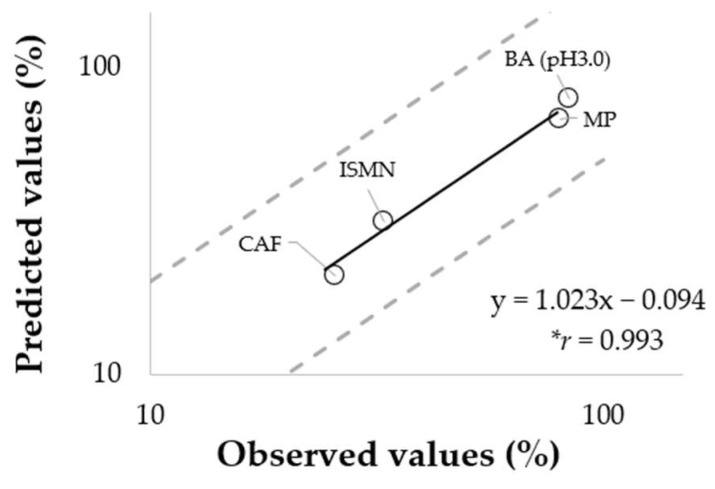
Comparison of predicted values using Strat-M^®^ versus observed values using porcine skin for percutaneous absorption ratio. Each point represents the mean, while dashed lines correspond to a factor 2. The Pearson’s correlation coefficient showed statistical significance (* *p* < 0.05).

**Figure 7 pharmaceutics-14-01370-f007:**
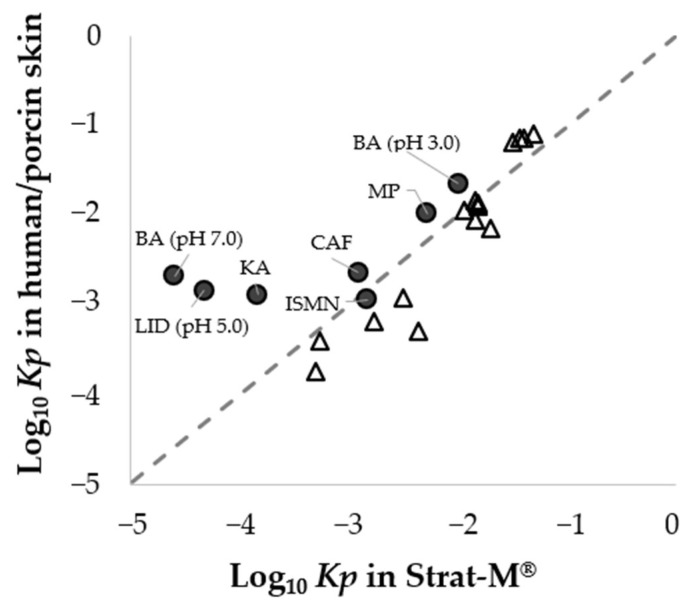
Comparison of *Kp* values between our research (porcine skin) and the published research (human skin). Circles (●) represent our dataset (−2.3 ≤ Log *K_ow_* ≤ 2.0), while unfilled triangles (Δ) represent the published dataset (−0.9 ≤ Log *K_ow_* ≤ 3.5, Uchida et al., 2015), and the dashed line corresponds to factor 1. Each point represents the mean (*n* = 3−4).

**Table 1 pharmaceutics-14-01370-t001:** Physicochemical properties of the examined target chemicals.

Chemicals	Abbreviations	*MW*	Log *K_ow_*	*Cv* (µg/mL)	Solvent
Kojic acid	KA	142.11	−0.9	1.04 × 10^4^	Water
Caffeine	CAF	194.19	−0.1	1.00 × 10^4^	Water
Isosorbide 5-mononitrate	ISMN	191.13	−0.2	1.10 × 10^4^	Water
Lidocaine	LID (pH 5.0)	234.34	−0.9	9.56 × 10^3^	pH 5.0 citrate buffer
Sodium benzoate	BA (pH 3.0)	122.12	1.9	1.83 × 10^3^	pH 3.0 citrate buffer
	BA (pH 7.0)	121.12	−2.3	1.08 × 10^4^	pH 7.0 PBS
Methyl paraben	MP	152.15	2.0	1.39 × 10^3^	Water

**Table 2 pharmaceutics-14-01370-t002:** Permeation parameters of chemicals assessed using porcine skin or Strat-M^®^.

Chemicals	Porcine Skin		Strat-M^®^	
	*Kp* (cm/h)	*K* (−)	*Kp* (cm/h)	*K* (−)
KA	(1.30 ± 0.37) × 10^−3^	(1.08 ± 0.15) × 10^−1^	(1.46 ± 0.30) × 10^−4^	(5.77 ± 2.32) × 10^−2^
CAF	(2.32 ± 0.12) × 10^−3^	(1.36 ± 0.62) × 10^−1^	(1.23 ± 0.26) × 10^−3^	(4.22 ± 0.91) × 10^−1^
ISMN	(1.15 ± 0.32) × 10^−3^	(1.01 ± 0.26) × 10^−1^	(1.49 ± 0.48) × 10^−3^	1.04 ± 0.37
LID (pH5.0)	(1.44 ± 0.69) × 10^−3^	(1.25 ± 0.83) × 10^−1^	(4.72 ± 2.43) × 10^−5^	(2.15 ± 1.71) × 10^−2^
BA (pH3.0)	(2.27 ± 0.07) × 10^−2^	1.51 ± 0.09	(1.01 ± 0.03) × 10^−2^	1.57 ± 0.20
BA (pH7.0)	(2.11 ± 2.10) × 10^−3^	(1.44 ± 1.31) × 10^−1^	(2.51 ± 0.25) × 10^−5^	(6.73 ± 1.04) × 10^−3^
MP	(1.06 ± 0.15) × 10^−2^	(7.08 ± 2.90) × 10^−1^	(5.20 ± 1.08) × 10^−3^	3.08 ± 0.65

Permeation parameters were calculated based on 4 data points between 2 h and 8 h in porcine skin or 3 data points between 4 h and 8 h in Strat-M^®^. Mean ± standard error (S.E.) (*n* = 3–4).

## Data Availability

The data presented in this study are available on request from the corresponding author.

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
