# Peer review of "A Mathematical Approach Using Strat-M® to Predict the Percutaneous Absorption of Chemicals under Finite Dose Conditions"

_pharmaceutics, 2022, doi:10.3390/pharmaceutics14071370_

Round 1
Reviewer 1 Report
A mathematical approach of skin absorption data in real condition of use (i.e. finite dose) is crucial for all people involved on this topic. Thanks to the authors to address this question. Moreover, alternative to costly ex vivo skin absorption assay is also of importance.
These aspects of the percutaneous absorption are simultaneously address in this paper.
See word document for comments

Reviewer 2 Report
The manuscript submitted for publication deals with the development of a mathematical model that included the permeation parameters as well as water evaporation to evaluate percutaneous absorption of different chemicals present in formulations for skin application.
The selected topic is pertinent, and the structure and the content of the manuscript are convincing with valuable and consistent findings. Furthermore, the paper is well-written.
The rationale behind the selection of chemicals used in the different experiments should be improved, as two of them, 5-isosorbide mononitrate and lidocaine, are not cosmetic ingredients, but should be included in pharmaceutical formulations for skin application. Therefore, the title must be revised and throughout the manuscript, it should be changed so that it could be applied to both cosmetic and dermopharmaceutical products.
Minor point:
- Check the meaning of the sentence “Each sample was inserted into the (Nexera-i LC-2040C plus HPLC system (Shimadzu Co., Kyoto, Japan)” (page 4, lines 164-165)
With thanks and best wishes,
The Reviewer
Round 2
Reviewer 1 Report
All comments were adresses by the authors.
I still have three minor comments.
Equation 9 : Please check the equation 9 and also previous equation. Thecorrective factor VscK/(V+VscK) should be replaced by VscK/(V+Vsc) according to equations 6 to 8.
Point 5) Equation 4 : I agree. As aqueous solution was used, you can assume that partition and diffusion coefficient should not depend on vehicle evaporation. This assumption has to be done to simplify data treatment. Nevertheless, it’s not so obvious. Indeed, as water evaporates, chemical concentration increases until the saturation, then the chemical should precipitate.
Point 8) Thanks for the clarification. As you have in mind to calculate the amount of chemical which permeates below the SC, thickness L should represent SC thickness. Otherwise, you would have a contradiction between the equations describing chemical permeation through the skin and parameters used in these equations. In contrary, on StratM, it’s impossible to define the thickness of SC equivalent. Thus, to have consistent treatment between StratM and skin, you used thickness membrane to define L. Nevertheless, you have to deal with the contradiction.
From my personal point of view, I prefer to Sc thickness rather than membrane thickness. It’s more logical from a mathematical point of view.
